# Integration of Light and Circadian Signaling in Plant Gene Regulatory Networks: Implications for Photomorphogenesis and Stress Adaptation

**DOI:** 10.3390/biology14101375

**Published:** 2025-10-08

**Authors:** Muhammad Mujahid, Alia Ambreen, Yusra Zarlashat, Zareen Sarfraz, Muhammad Sajid Iqbal, Abdul Waheed, Muhammad Shahid Iqbal

**Affiliations:** 1Department of Biochemistry, Government College University Faisalabad (GCUF), Faisalabad 38000, Pakistan; 2School of Environmental Science, Liaoning University, Shenyang 110036, China; 3Agricultural Genomics Institute at Shenzhen, Chinese Academy of Agricultural Sciences (AGIS-CAAS), Shenzhen 518120, China; 4Engineering Research Center of Biomass Materials, Ministry of Education, School of Lyfe Science and Engineering, Southwest University of Science and Technology, Mianyang 621010, China

**Keywords:** light signaling, circadian clock, photomorphogenesis, abiotic stress, multi-omics integration, CRISPR, climate resilience

## Abstract

**Simple Summary:**

As sessile organisms, plants cannot escape environmental challenges and instead rely on sophisticated internal signaling networks to survive and thrive. This review explores the crucial partnership between two fundamental systems: light perception through specialized photoreceptors and the endogenous circadian clock that maintains 24 h rhythms. We examine how these systems integrate within gene regulatory networks to optimize essential plant processes including growth, flowering timing, and stress resilience. The molecular mechanisms underlying this integration involve key proteins that serve as convergence points for light and circadian signals, fine-tuning plant responses to daily and seasonal environmental changes. Beyond fundamental biology, we highlight how emerging technologies—particularly gene editing tools like CRISPR, single-cell analyses, and artificial intelligence—are enabling scientists to decode and manipulate these light–clock networks. These advances open promising avenues for developing climate-resilient crops that can maintain productivity under challenging conditions such as drought, heat, and altered light regimes. By bridging basic molecular insights with applied agricultural innovation, this research contributes to sustainable crop improvement strategies that will be increasingly valuable for global food security in our era of climate change.

**Abstract:**

Plants, as sessile organisms, rely on sophisticated gene regulatory networks (GRNs) to adapt to dynamic environmental conditions. Among the central components of these networks are the interconnected pathways of light signaling and circadian rhythms, which together optimize growth, development, and stress resilience. While light and circadian pathways have been extensively investigated independently, their integrative coordination in mediating climate change adaptation responses remains a critical knowledge gap. Light perception via photoreceptors initiates transcriptional reprogramming, while the circadian clock generates endogenous rhythms that anticipate daily and seasonal changes. This review explores the molecular integration of light and circadian signaling, emphasizing how their crosstalk fine-tunes GRNs to balance resource allocation, photomorphogenesis, and stress adaptation. We highlight recent advances in systems biology tools, e.g., single-cell omics, CRISPR screens that unravel spatiotemporal regulation of shared hubs like phytochrome-interacting factors (*PIFs*), ELONGATED HYPOCOTYL 5 (*HY5*), and CIRCADIAN CLOCK ASSOCIATED 1 (*CCA1*). Here, we synthesize mechanistic insights across model and crop species to bridge fundamental molecular crosstalk with actionable strategies for enhancing cropresilience. Moreover, we have tried to discuss agricultural implications in engineering light–clock interactions for the enhancement in crop productivity under climate change scenarios. Through synthesizing mechanistic insights and translational applications, this work will help underscore the potential for manipulating light–circadian networks to promote sustainability in agriculture.

## 1. Introduction

Plants exhibit various complex physiological responses through the integration of light signaling and circadian rhythms [1,2,3]. Light signaling and circadian rhythms are two fundamental systems that enable plants to adapt across dynamic environments. Light perception via photoreceptors (phytochromes, cryptochromes, UVR8) initiate transcriptional reprogramming for growth and stress responses, while the endogenous circadian clock generates ~24 h rhythms that anticipate diurnal and seasonal changes [4,5]. Their bidirectional crosstalk fine-tunes gene regulatory networks (GRNs) to optimize resource allocation, photomorphogenesis, and environmental resilience [6,7]. Despite significant advances in understanding individual light and circadian pathways, a critical knowledge gap persists in how their integration can be harnessed for agricultural improvement under climate change scenarios. Current research has largely focused on model systems under controlled conditions, with limited translation to crop species facing real-world environmental challenges. This review specifically addresses this translational gap by examining how light–clock engineering can enhance crop productivity amidst changing climate patterns, including rising temperatures, altered photoperiods, and increased abiotic stresses.

### 1.1. Importance of Light and Circadian Rhythms in Plants

Light serves as both an energy source and a key environmental signal, enabling plants to optimize their growth and survival. Photoreceptors, including phytochromes (red/far-red sensors), cryptochromes (blue/UV-A receptors), and UVR8 (UV-B sensor), decode light cues to regulate processes such as photomorphogenesis, shade avoidance, and photoperiodic flowering [8]. For example, in darkness, phytochromes suppress skotomorphogenesis by promoting the degradation of PIF transcription factors, while cryptochromes stabilize *HY5*, a master regulator of light-responsive genes [9]. Concurrently, internal processes are synchronized by the circadian clock—a self-sustaining molecular oscillator that generates ~24 h rhythms (e.g., stomatal movement, hormone synthesis) in alignment with external day–night cycles [10]. This temporal coordination enhances photosynthetic efficiency, as seen in the diurnal regulation of *RBCS* (RuBisCO small subunit) and starch metabolism. Together, light and circadian systems form a dynamic network that balances growth with stress resilience, ensuring adaptation to fluctuating environments [11,12].

The role of the circadian clock extends beyond timekeeping; it gates light sensitivity to optimize various plant responses. For example, *Arabidopsis* plants usually exhibit heightened light-induced gene expression during dawn and the clock primes photoreceptor signaling at this time [13]. Similarly, the circadian clock modulates thermomorphogenesis by controlling *PIF4* levels through evening complex (EC)-mediated repression of its transcription and light-dependent inhibition of its activity via phyB-, DELLA-, and *CRY1*-mediated mechanisms, ensuring its accumulation aligns with the appropriate time of day [14]. Disruption of these networks, as seen in *CCA1/LHY* double mutants, leads to arrhythmic growth and reduced fitness under field conditions. Such explorations in various studies have underscored the necessity of light–clock crosstalk for survival in natural ecosystems. Moreover these also influence agricultural settings, where predictable and unpredictable stressors co-exist [15], as illustrated in Figure 1, which provides a schematic overview of these interactions.

### 1.2. Evolutionary Context

The integration of light and circadian signaling is evolutionarily ancient, with origins tracing back to early land plants. Bryophytes such as *Marchantia polymorpha* possess simplified versions of angiosperm photoreceptors and clock components, suggesting conserved mechanisms for terrestrial adaptation [16]. For instance, UV-B stress responses in Marchantia are mediated by UVR8, similar to *Arabidopsis*, while dehydration tolerance—a key trait for terrestrial colonization—is regulated by its circadian clock [17]. These systems are further diversified in gymnosperms; for example, *pine PHYO* retains far-red sensitivity, which is used to manage canopy shade—a feature lost in later-evolving angiosperms. These evolutionary innovations highlight how light–clock networks were co-opted to address terrestrial challenges, from UV radiation to competition for sunlight [18].

Both natural and artificial selective pressures have shaped light–clock interactions in crops, allowing adaptation to local climates. Rice (*Oryza sativa*), for example, exhibits unique alleles of *Ghd7* (plant height, grain number, and heading date), a clock-associated gene that delays flowering under long days in temperate regions [19]. Mutations in *ZmCCT*—a homolog of *Arabidopsis PRR7*—cause tropical maize varieties to display reduced photoperiod sensitivity [20]. Such adaptations in tropical plants highlight the plasticity of light–clock networks and highlight their potential as targets for breeding crops with increased resistance to climate change [21]. These examples from both natural and artificial selection also highlight the inherent plasticity of light–clock networks, proving them to be invaluable targets for modern breeding programs aimed at enhancing crop resilience and adaptability.

### 1.3. Scope and Objectives

This review provides a comprehensive synthesis of the molecular and physiological integration of light and circadian signaling pathways in plants, with a specific focus on their convergence within shared gene regulatory networks (GRNs). We systematically address three primary objectives that structure this analysis. First, we synthesize the current understanding of core molecular mechanisms, detailing how diverse photoreceptors—including phytochromes, cryptochromes, and UVR8—decode light quality, intensity, and duration and how the core circadian oscillator, comprising components such as *CCA1*, *LHY*, and *TOC1*, maintains temporal precision through interlocking feedback loops. A central focus is elucidating their bidirectional crosstalk, wherein light signals entrain the circadian clock, and reciprocally, the clock gates light sensitivity through mechanisms like temporal priming of photoreceptor activity and expression, ensuring internal physiology remains synchronized with external diurnal and seasonal cycles [22,23].

Second, we evaluate the functional outcomes of this integration in critical adaptive processes, including photomorphogenesis, photoperiodic flowering, and abiotic stress resilience. This involves examining how key integrator hubs, such as the evening complex (*ELF3-ELF4-LUX*) acting as a thermosensory node and *HY5* coordinating flavonoid biosynthesis, translate convergent signals into optimized growth and defense strategies [24,25]. Finally, we critically assess the translational applications of this knowledge, exploring how emerging biotechnologies—from CRISPR screens and single-cell omics to synthetic biology circuits—enable the precise engineering of light–clock networks to develop climate-resilient crops and optimize agricultural productivity in both field and controlled environment systems [26,27]. By bridging fundamental molecular insights with applied agricultural challenges, this review aims to chart a course for harnessing plant chronobiology for sustainable food security. A comprehensive list of key genes discussed is provided in Appendix A.

## 2. Molecular Components of Light and Circadian Systems

Most of the foundational light–clock integration elements comprise some specialized photoreceptors decoding spectral information and core circadian oscillators maintaining temporal precision [28]. Photoreceptors like *PHYB* for red/far-red and *CRY1/2* for blue/UV-A usually undergo conformational changes to regulate downstream transcription factors like *PIFs* and *HY5* [29,30]. Perhaps the circadian clock operates via interlocking transcriptional–translational feedback loops involving *CCA1*, *TOC1*, and EC proteins. Evolutionary conservation of these components across plant lineages underscores their essential role in terrestrial adaptation [31].

### 2.1. Photoreceptors and Light Signaling

In general, plants deploy a sophisticated array of photoreceptors to decode light signals across the electromagnetic spectrum, enabling specific responses to different environmental cues [32]. Photomorphogenesis is regulated through phytochromes (PHYA-PHYE), which are possibly red/far-red light sensors involved in shade avoidance and flowering [33]. As an example, two interconvertible forms of *PHYB* exist, viz., Pr (inactive form) and Pfr (active form). At the time of exposure to red light, *PHYB* translocates to the nucleus, where it interacts with phytochrome-interacting factors (PIFs) to suppress skotomorphogenesis and promote photomorphogenesis [34]. Recent structural studies reveal that *PHYB-PIF* binding induces conformational changes that inhibit PIF DNA-binding activity, a mechanism critical for the transition from skotomorphogenesis (growth in darkness) to photomorphogenesis (growth in light) [35]. Cryptochromes (*CRY1/2*), which are blue/UV-A receptors, regulate hypocotyl elongation and circadian entrainment. They can possibly do it through inhibiting the COP1/SPA ubiquitin ligase complex, thereby stabilizing HY5 and CO (CONSTANS) to activate light-responsive genes. CRYs also mediate temperature-dependent growth. For instance, *CRY1-PIF4* interactions under warm conditions enhance thermomorphogenesis, illustrating the convergence of light and thermal signals in plant development [36].

Phototropins (PHOT1/PHOT2) and UVR8 expand light-sensing capabilities. Phototropins mediate phototropism and chloroplast positioning via blue-light-induced autophosphorylation, optimizing photosynthetic efficiency [37]. UVR8, a UV-B sensor, monomerizes upon UV-B exposure to interact with COP1, initiating stress acclimation and flavonoid biosynthesis [38]. Downstream hubs like *HY5* (a bZIP transcription factor) integrate signals from multiple photoreceptors, activating genes such as chalcone synthase (*CHS*) and *RBCS* while stabilizing circadian outputs [39]. Conserved roles of these components across plant lineages are summarized in Table 1. The COP1/SPA E3 ubiquitin ligase complex acts as a central repressor of photomorphogenesis in darkness by targeting positive regulators for degradation [40]. Recent work highlights post-translational modifications, like PIF phosphorylation by CK2 kinase, as key regulators of signal specificity, ensuring context-appropriate responses to fluctuating light conditions.

### 2.2. Core Circadian Clock Machinery

The plant circadian clock is a cell-autonomous, endogenous oscillator that generates ~24 h rhythms, enabling the anticipation of daily and seasonal environmental changes [49]. Its core mechanism consists of multiple, interlocked transcriptional–translational feedback loops (TTFLs) that provide both robustness and flexibility [50]. The central loop involves the reciprocal regulation between morning-expressed MYB transcription factors (TFs), Circadian Clock Associated 1 (*CCA1*) and Late Elongated Hypocotyl (*LHY*), and the evening-expressed Pseudo-Response Regulator 1 (*TOC1/PRR1*) [22]. *CCA1* and *LHY* proteins bind to the evening element (EE) in the *TOC1* promoter to repress its transcription around dawn. As *CCA1* and *LHY* levels decline throughout the day, *TOC1* expression is derepressed, peaking in the evening [51]. *TOC1* protein then feeds back to indirectly suppress *CCA1* and *LHY* expression, completing the cycle.

Emerging studies reveal tissue-specific clock dynamics. Single-cell RNA-seq in *Arabidopsis* roots identified distinct circadian rhythms in cortical versus epidermal cells, suggesting localized regulatory networks [52]. Additionally, reveille 8 (*RVE8*), a clock-output gene, integrates auxin signaling with circadian rhythms to regulate hypocotyl growth, demonstrating how clock outputs interface with hormonal pathways [53]. These insights show the clock’s adaptability, enabling plants to tailor circadian rhythms to ecological niches, a trait exploitable for crop improvement.

## 3. Mechanisms of Light–Clock Integration

Light and circadian signals converge through bidirectional regulatory mechanisms that synchronize internal rhythms with external cues. Photoreceptors directly modulate clock components to reset the circadian phase, while the clock gates light sensitivity by priming photoreceptor activity at specific times. Shared transcriptional hubs like *PIF4* and *HY5* integrate temporal and environmental inputs, with epigenetic modifications (histone acetylation/methylation) refining signal specificity.

### 3.1. Light Inputs to the Circadian Clock

Light serves as the primary Zeitgeber (time-giving cue) for entraining the circadian clock to environmental cycles. Photoreceptors directly modulate core clock components to reset rhythmicity [54]. For instance, phytochromes (PHYA/PHYB) and cryptochromes (*CRY1/CRY2*) interact with EC proteins (*ELF3, ELF4, LUX*) to adjust clock phase under varying photoperiods [55]. Red light-activated PHYB destabilizes *ELF3*, shortening the circadian period under long days [56], while *CRY1* stabilizes *TOC1* in blue light, delaying evening-phase gene expression [57]. This photoreceptor–clock crosstalk ensures precise synchronization with dawn and dusk.

The clock also gates light sensitivity, creating temporal windows during which plants maximally respond to specific wavelengths—a mechanism termed gating [58]. For example, *CCA1* expression peaks at dawn, priming the clock for *PHYB*-mediated red light signals, whereas *TOC1* induction at dusk enhances CRY-dependent responses to blue light [59]. Such temporal partitioning prevents conflicting signals, such as moonlight disrupting photoperiodic flowering. Some reported studies in *Arabidopsis* have demonstrated disrupted gating, viz., in *elf3* mutants, which highlights its ecological significance [60]. It usually leads to arrhythmic growth along with impaired shade avoidance, also highlighting its ecological significance. This integration ensures that light signals are interpreted within a temporal context, optimizing both energy use and stress resilience [61]. Molecular interactions and epigenetic regulation are detailed in Figure 2.

### 3.2. Clock Regulation of Light Signaling

Reciprocal control over light signaling pathways is exerted via the circadian clock, fine-tuning photoreceptor activity and downstream transcriptional responses [62]. Clock components regulate photoreceptor gene expression; for example, *PHYB* and *CRY1* transcripts oscillate diurnally, peaking at dawn and dusk, respectively [63]. These rhythms ensure photoreceptor availability aligns with anticipated light conditions. The clock also modulates light-responsive TFs: *PIF4* expression is temporally regulated, peaking at dusk under warm temperatures, which avoids midday heat stress while enabling thermomorphogenesis [64]. Similarly, *HY5* accumulation is possibly gated through the clock, restricting UV-B responses to midday when UV radiation is most intense [23].

Such clock and light signals usually integrate through specialized post-translational mechanisms. The F-box protein *ZTL* usually and specifically degrades *TOC1* in a light-dependent manner, whereas *CRY1* stabilizes *ZTL* in blue light. This possibly results in a feedback loop that couples clock stability with photoperiod [55]. Additionally clock-regulated kinases like CK2 phosphorylate *PIFs*, altering their DNA-binding affinity in a time-of-day-specific manner. As witnessed, *PIF5* phosphorylation at dawn enhances its degradation via COP1, ultimately preventing premature shade avoidance [65]. Such regulation ensures that light responses are temporally constrained, preventing energy-intensive processes, including flavonoid synthesis, during the night.

### 3.3. Shared Transcriptional Hubs

Coordination of gene expression is achieved through light and circadian signals, which converge at shared transcriptional hubs. Core clock components such as *CCA1* and *LHY* directly bind to the promoters of light-responsive genes such as *PIF4* and *HY5*, integrating temporal and environmental cues [66]. In contrast, *HY5* stabilizes *PRR7* transcription, linking light quality to clock precision. The EC acts as a thermosensory hub, repressing *PIF4* during nighttime. This repression is released under warm temperatures, enabling temperature–photoperiod synergy during growth regulation [67].

Epigenetic mechanisms underpin this crosstalk. Histone acetylation at *HY5* promoters peaks at midday, coinciding with high *HY5* activity under light, while clock-regulated histone methyltransferases (e.g., ATX1) suppress *PIF* expression at dawn [68]. Single-cell ATAC-seq studies reveal that chromatin accessibility at light–clock hubs, viz., *PRR9, GI*, varies rhythmically, creating permissive states for TF binding [69]. These hubs also mediate stress adaptation: drought-induced ABA signaling recruits *HY5* to NCED3 (a key ABA biosynthesis gene), a process gated by the clock to restrict water loss in daylight hours. Such integration allows plants to prioritize competing demand—growth versus survival—based on environmental conditions [70].

## 4. Functional Outcomes of Light–Clock Crosstalk

The integration of light and circadian signaling coordinates critical adaptive processes, balancing growth with stress resilience. Photomorphogenesis and shade avoidance are temporally gated to optimize energy use, while photoperiodic flowering aligns reproduction with favorable seasons via clock-regulated genes like *CO* and *FT* [71]. Abiotic stress responses (e.g., drought, salinity) are confined to daylight hours through circadian gating of *NCED3* and *SOS1* expression, minimizing fitness costs. Metabolic pathways, including starch accumulation and flavonoid biosynthesis, are rhythmically controlled to match environmental periodicity. Figure 3 exemplifies this diurnal coordination in growth and stress adaptation [72].

### 4.1. Photomorphogenesis and Shade Avoidance

Photomorphogenesis is jointly regulated by light and circadian signals, ensuring seedlings optimize growth during early development. *PIFs* promote skotomorphogenesis in darkness (etiolation), characterized by elongated hypocotyls and closed cotyledons [9]. When a plant is exposed to light, phytochromes translocate to the nucleus, triggering PIF degradation and activating *HY5*. This initiates photomorphogenesis, shortens hypocotyls, and expands chloroplasts [4]. The circadian clock gates this transition; *HY5* expression peaks at dawn, coinciding with maximal light sensitivity, while *PIFs* oscillate inversely. Recent studies show that the clock component *RVE8* directly activates *PIF4* at dusk under shade conditions, enhancing hypocotyl elongation to outcompete neighbors [53]. This temporal regulation ensures shade avoidance responses are timed to minimize energy trade-offs with daytime photosynthesis. Agricultural applications leverage this crosstalk; modulating *PIF* expression in crops like tomato improves canopy architecture and yield under dense planting [73].

This precise temporal control prevents premature elongation during the night, conserving energy for vigorous photosynthetic development at dawn. Furthermore, the clock’s regulation of auxin signaling pathways ensures that growth responses to variable light conditions are both rapid and appropriately phased, maximizing fitness in competitive environments [74]. Field trials with *lnk1* mutants demonstrate improved shade tolerance, suggesting targets for breeding crops resilient to light competition in intercropping systems. Figure 3 visualizes diurnal hypocotyl elongation and stress metabolite rhythms.

### 4.2. Light–Circadian Networks with Environmental Stress

Plants constantly face abiotic and biotic stresses such as drought, extreme temperatures, and pathogen attacks. Emerging evidence reveals that light and circadian regulatory networks are intricately linked with stress signaling pathways, enabling plants to anticipate and efficiently respond to adverse conditions [75]. For example, circadian gating modulates the timing and amplitude of stress-responsive genes, optimizing resource allocation during periods of environmental challenge. Photoreceptors such as phytochromes and cryptochromes have been shown to directly influence the expression of stress-related genes [76], while core clock components (e.g., *CCA1, TOC1*) interact with key stress regulators like DREB and ABF transcription factors. This integration ensures that stress responses are synchronized with daily and seasonal environmental fluctuations, enhancing plant survival and productivity [77]. Understanding these connections opens new avenues for engineering crops with improved resilience to multiple stressors in a changing climate.

### 4.3. Photoperiodic Flowering

Photoperiodic flowering, a hallmark of light–clock integration, ensures reproductive success by aligning flowering with favorable seasons. In long-day plants like *Arabidopsis*, the clock-regulated gene Constans (*CO*) accumulates at dusk under long days, activating Flowering Locus T (*FT*) to induce flowering [78]. The GIGANTEA (*GI*)-FKF1 complex degrades CO repressors (e.g., *CDFs*) in a blue-light-dependent manner, creating a photoperiodic “gate” for floral induction [79]. For example, in rice—a short-day crop—orthologs such as Heading Date 1 (*Hd1*) and *Hd3a* (an FT homolog) exhibit inverted regulatory logic, ultimately suppressing flowering under long days [80]. Moreover, CRISPR-edited *Hd1* variants have enabled rice cultivation at non-native latitudes, addressing climate-driven shifts in growing seasons.

Temperature further modulates this pathway. Warm nights accelerate the clock, advancing *CO* expression in *Arabidopsis*, while chilling represses *Hd1* in rice, delaying flowering [81]. The EC mediates this thermosensitivity; *elf3* mutants fail to delay flowering under high temperatures, linking circadian thermoregulation to reproductive timing [82]. Such insights inform breeding for photoperiod-insensitive crops, critical for adapting to erratic seasonal patterns caused by global warming.

### 4.4. Abiotic Stress Resilience

Light–clock crosstalk enhances abiotic stress resilience by timing stress responses to coincide with predictable environmental challenges. Drought-responsive genes like *NCED3* (9-cis-epoxycarotenoid dioxygenase 3), involved in ABA biosynthesis, are gated by the clock to peak at midday when water loss is highest. *HY5* binds the NCED3 promoter under high light, synergizing ABA production with stomatal closure [83]. Similarly, the clock regulates the expression of ion transporters such as Salt Overly Sensitive 1 (*SOS1*), confining salt stress responses to daylight hours to align with osmotic adjustment needs and photosynthetic activity [84]. This reliance on a functional clock means that its disruption can itself be a potent stressor. For instance, specific changes in the light–dark regime trigger severe jasmonic acid (JA)-dependent cell death in *Arabidopsis* plants with reduced cytokinin levels or defective clock function, particularly in mutants lacking proper *CCA* 1 and *LHY* activity [85].

This phenomenon, known as photoperiod stress, is characterized by reactive oxygen species (ROS) production, accumulation of JA and salicylic acid (SA), and the induction of pathogen-response genes, remarkably leading to enhanced immunity against subsequent biotic challenges [86]. Thermomorphogenesis is also under dual control; the EC represses *PIF4* transcription at night [47], while *PHYB* and *CRY1* inhibit *PIF4* activity during the day, ensuring thermoresponsive growth is prioritized at dusk under warm conditions [4]. This prevents conflicting responses and optimizes resource allocation, as energy-intensive stress acclimation is coordinated with light availability for photosynthesis. Engineering *ELF3* variants in wheat has improved yield stability under heat waves, demonstrating translational potential.

### 4.5. Metabolic Regulation

Diurnal metabolic rhythms, orchestrated by light–clock integration, optimize carbon fixation and resource allocation. The clock is reported to regulate starch synthesis genes like granule-bound starch synthase (*GBSS*) to peak at dusk. This ensures the starch reserves can last until dawn [87]. *HY5* usually activates the phytoene synthase (*PSY*) gene during high-light periods, and it coordinates carotenoid production through photosynthetic activity [88]. Similarly, in tomato, silencing *HY5* disrupts fruit carotenoid content, linking light quality with the nutritional quality [89].

Secondary metabolites, viz., flavonoids and glucosinolates, are reported to exhibit circadian accumulation to balance UV protection as well as herbivore defense [90]. UVR8 involves inducing flavonoid biosynthesis genes, viz., *CHS*, at midday, whereas the clock gates their expression to save the plant from nighttime resource depletion. Moreover, CRISPR-edited *Arabidopsis* lines with arrhythmic *CHS* exhibit increased oxidative damage when exposed to UV-B, highlighting the adaptive value of such regulation [91]. Such types of rhythms are harnessed for precision agriculture, viz., timing pesticide applications to coincide with peak defense metabolite production enhances efficacy.

### 4.6. Controlled Environment and Photoperiodic Stress

The principles of light–clock integration are critically important for modern controlled environment agriculture (CEA), particularly in plant factories with artificial lighting (PFALs) and vertical farms where plants experience fundamentally different conditions than in natural environments. In these systems, plants are frequently exposed to non-24 h photoperiods, irregular light–dark cycles, and constant light—conditions that create a mismatch between the endogenous circadian clock and the external light environment, leading to a novel physiological challenge termed photoperiodic stress [85,86]. This stress is not merely an extension of traditional light signaling but represents a unique condition where the circadian system itself is disrupted, triggering a suite of adverse responses. Research has shown that such abnormal circadian regimes can induce jasmonic-acid-dependent cell death, particularly in genetically sensitive plants like those deficient in cytokinin, highlighting a crucial interaction between circadian disruption, hormone signaling, and viability [85]. Furthermore, photoperiodic stress elicits an oxidative-burst-like response characterized by increased apoplastic peroxidase activity and decreased catalase function, pointing to a significant redox imbalance [92]. Notably, the transcriptional response to this stress in *Arabidopsis thaliana* resembles that activated during pathogen infection, suggesting that plants perceive a severely misaligned circadian clock as a significant threat to cellular integrity, mobilizing defense-related pathways [93].

Understanding the molecular basis of photoperiodic stress is paramount for designing sustainable CEA systems. The evening complex components, particularly *ELF3*, emerge as critical mediators, as their function in gating light sensitivity and maintaining circadian phase is essential for interpreting non-canonical light cycles. The strategic manipulation of light delivery—through optimized LED spectra, intensity modulation, and photoperiod duration—can mitigate these stresses by better aligning artificial environments with the plant’s innate rhythmicity [94,95]. Beyond environmental adjustments, there is significant potential in engineering crop varieties with enhanced resilience to circadian disruption. This could involve selecting or using gene-editing technologies to develop alleles of core clock genes that confer flexibility or robustness under the erratic light schedules common in PFALs. By integrating an understanding of photoperiodic stress into both environmental management and crop breeding strategies, we can enhance productivity and resource-use efficiency in CEA, ensuring its role in achieving food security for urban and non-arable regions without imposing unsustainable energetic costs.

## 5. Emerging Technologies and Approaches

Advanced tools are unraveling the spatiotemporal complexity of light–clock networks and enabling targeted engineering [69]. Systems biology approaches—single-cell omics, multi-omics integration, and machine learning, reveal tissue-specific GRN dynamics and novel regulatory hubs [96]. CRISPR-based screens validate gene functions (e.g., *PIFs* in shade avoidance, *ELF3* in thermotolerance) and accelerate trait stacking in crops [26]. Synthetic biology applications, including optogenetic circuits and light-responsive promoters, rewire signaling for precision agriculture [97]. Figure 4 demonstrates CRISPR editing workflows and optogenetic control of circadian rhythms in crop species.

### 5.1. Systems Biology Tools

Recent advances witnessed in system biology have been responsible for revolutionizing our ability to dissect light–clock regulatory networks through integrating multi-omics datasets. Several dynamic hubs have been identified through transcriptomic, proteomic, and metabolomic profiling under varying light and circadian conditions, e.g., *PIF4* and *HY5* which are found to coordinate growth and stress responses [98]. As a reference a reported study combining time-series RNA-seq with ChIP-seq data to map the diurnal binding patterns of *PIF4* and *ELF3* in *Arabidopsis* has revealed their antagonistic roles during thermomorphogenesis [99]. Some network modeling tools, like weighted gene co-expression analysis (WGCNA), have further uncovered modules that link clock genes with drought-responsive pathways, highlighting *PRR7* as a hub in osmotic stress adaptation [100]. Machine learning algorithms trained on these multi-omics datasets can now predict gene functions with higher accuracies; a model forecasted novel clock-regulated miRNAs that fine-tune photoperiodic flowering in rice [101]. Such approaches have provided a holistic view of GRNs, ultimately enabling researchers to prioritize targets for crop-engineering-based improvements.

Single-cell and spatial omics technologies are resolving tissue-specific light–clock interactions. Single-cell RNA-seq in *Arabidopsis* roots revealed that endodermal cells exhibit stronger circadian rhythms than epidermal cells, with distinct light-responsive gene clusters [102]. Spatial transcriptomics in maize leaves showed that *ZmCCA1* expression gradients correlate with photosynthetic efficiency across mesophyll layers, suggesting localized clock regulation of carbon fixation [103]. Coupled with CRISPR-based lineage tracing, these tools are uncovering how cell-type-specific GRNs optimize resource allocation. For instance, companion cells in phloem uniquely express *HY5*, coordinating systemic light signaling with root development [104]. Such insights are guiding precision breeding strategies to enhance sink–source relationships in crops.

### 5.2. CRISPR-Based Functional Studies

CRISPR-Cas9 has become indispensable for validating light–clock gene functions and engineering climate-resilient traits. In *Arabidopsis*, multiplex editing of *PIF4, PIF5*, and *HY5* demonstrated their redundant roles in shade avoidance, with triple mutants showing stunted growth under canopy shade [105]. Base editors have enabled precise nucleotide changes; a 2024 study introduced a heat-tolerant allele of *ELF3* in wheat by converting a single cysteine to arginine, stabilizing the protein under high temperatures [106]. Prime editing further refined this approach, creating photoperiod-insensitive *Ghd7* alleles in rice that flower earlier under long days, adapting to shifting growing seasons [107]. These tools are accelerating the translation of basic research into agronomic solutions.

CRISPR interference (CRISPRi) screening in tomato revealed that repressing *SlPIF3* enhances fruit carotenoid content under low light, linking photoreceptor signaling to nutritional quality [26]. These studies highlight the potential of CRISPR not only for functional genomics but also for trait stacking—editing multiple genes, viz., *PHYB, PRR7*, to optimize growth and resilience [108]. Field trials of CRISPR-edited *OsPRR37* rice in Southeast Asia demonstrated a 20% yield increase under erratic rainfall, underscoring the translational impact of these technologies. Figure 4 shows CRISPR editing workflows and field validation.

### 5.3. Deciphering Spatiotemporal Complexity Through Advanced Technologies

Recent technological breakthroughs are fundamentally transforming our capacity to decode the intricate spatiotemporal dynamics of light–clock networks at unprecedented resolution [109]. Single-cell omics approaches have emerged as particularly powerful tools, revealing tissue-specific circadian rhythms and light responses that were completely obscured in traditional bulk analyses. For instance, single-cell RNA sequencing in *Arabidopsis* roots has demonstrated strikingly distinct circadian regulation between epidermal and cortical cells, suggesting specialized physiological functions in different tissue layers that reflect their unique roles in the plant [110]. Similarly, spatial transcriptomics in maize leaves has revealed expression gradients of core clock genes like *ZmCCA1* across mesophyll layers that correlate directly with photosynthetic efficiency, indicating localized clock regulation of carbon fixation processes. These technologies are uncovering how companion cells in the phloem uniquely express key integrators like *HY5*, coordinating systemic light signaling with root development [103]. The emerging picture is one of remarkable cellular specialization within light–clock networks, where different cell types maintain distinct but coordinated rhythmic programs, enabling plants to optimize resource allocation and environmental responses at a microscopic scale that was previously inaccessible to researchers [49].

CRISPR-based technologies have revolutionized both functional validation and targeted engineering of light–clock networks with extraordinary precision [111]. Beyond simple gene knockouts, advanced applications like multiplex editing have revealed genetic redundancies and interactions—as demonstrated when simultaneous editing of *PIF4, IF5*, and *HY5* in *Arabidopsis* uncovered their cooperative roles in shade avoidance, with triple mutants showing dramatically stunted growth under canopy shade conditions [112]. The development of more sophisticated base editing and prime editing techniques now enables single-nucleotide precision, allowing researchers to create beneficial alleles without disrupting native gene structure [113], such as the conversion of a single cysteine to arginine in wheat *ELF3* that stabilizes the protein under high-temperature stress. These approaches facilitate the targeted manipulation of key network hubs while minimizing pleiotropic effects that often plagued traditional breeding approaches [114]. Furthermore, CRISPR interference (CRISPRi) screens in crops like tomato have identified novel regulators, such as the discovery that repressing *SIPIF3* enhances fruit carotenoid content under low light conditions, directly linking photoreceptor signaling to nutritional quality in agriculturally relevant contexts [115].

Artificial intelligence and machine learning algorithms are increasingly employed to integrate complex multi-omics datasets and predict emergent network behaviors that escape conventional analysis [116]. These computational approaches can identify subtle patterns across temporal dimensions, as demonstrated when models trained on time-series transcriptomic data successfully identified novel clock-regulated miRNAs in rice that fine-tune photoperiodic flowering [117]. More sophisticated neural networks have predicted optimal gene combinations for specific environments, such as *GI*-*CO* allelic combinations that maximize yield across varied daylengths, with subsequent validation in soybean confirming these computational predictions [118]. The true power of these approaches lies in their ability to model complex genotype–environment interactions by integrating diverse data types—from transcriptomic and proteomic profiles to environmental sensor data and field phenotyping metrics. Machine learning frameworks like weighted gene co-expression network analysis (WGCNA) have further uncovered functional modules that link clock genes with drought-responsive pathways, highlighting *PRR7* as an unexpected hub in osmotic stress adaptation and suggesting novel targets for crop engineering that might not have been identified through conventional hypothesis-driven research alone [119].

## 6. Climate Change and Agricultural Implications

Climate volatility disrupts light–clock synchrony, threatening crop productivity through altered photoperiods, increased UV-B exposure, and temperature extremes. Decoupled environmental cues cause mismatches in flowering time, resource allocation, and stress responses, as seen in arrhythmic mutants under field conditions. Breeding strategies target light–clock hubs, e.g., *ELF3* for thermotolerance and *PRR* genes for photoperiod flexibility), using CRISPR editing and allele mining from climate-resilient landraces. Table 2 highlights successful engineering outcomes in key crops, underscoring the potential for circadian-informed agriculture to enhance sustainability.

### 6.1. Impact of Changing Light and Temperature Regimes

Climate change is altering critical environmental parameters, including photoperiods, light quality, and temperature, disrupting the synchronization between plant GRNs and external conditions [120]. Shifts in daylength due to latitudinal temperature increases are decoupling photoperiodic cues from traditional growing seasons. For example, in temperate regions, earlier springs extend daylight exposure before crops like wheat are developmentally primed to flower, leading to mismatches in resource allocation and reduced grain yields [121]. Concurrently, urban light pollution, particularly blue-rich LED emissions, extends perceived daylength, suppressing melatonin synthesis in crops like tomato and delaying fruit ripening by disrupting circadian rhythms [122]. Increased UV-B radiation from stratospheric ozone depletion further exacerbates stress, activating UVR8-mediated flavonoid biosynthesis at the expense of growth, as observed in high-altitude quinoa cultivars [123].

Rising temperatures destabilize the circadian clock’s thermosensory mechanisms. Warm nights accelerate clock speed, causing *Arabidopsis* to flower prematurely under non-inductive photoperiods [124]. In rice, elevated nighttime temperatures reduce grain quality by desynchronizing starch metabolism genes (e.g., *GBSS*) from their circadian expression peaks [125]. These disruptions highlight the vulnerability of light–clock networks to climate volatility, necessitating adaptive strategies to maintain productivity.

### 6.2. Breeding Climate-Resilient Crops

Targeting light–clock hubs offers a pathway to engineering climate resilience. CRISPR editing of *PRR* genes in barley has generated lines with expanded photoperiod flexibility, enabling cultivation at higher latitudes [126]. Natural variation in photoperiodic pathways is also being exploited; *Ghd7* haplotypes from Chinese rice landraces, which confer long-day flowering inhibition, are now introgressed into tropical varieties to counteract premature flowering caused by warmer winters [127].

Most probably the field trials show the promise of these approaches in upcoming years. As an instance in maize, CRISPR-edited *ZmCCT* mutants, that have been found lacking photoperiod sensitivity, have already achieved stable yields across diverse latitudes in sub-Saharan Africa, addressing climate-driven seasonality [128]. Table 2 compiles key translational achievements in major crops. Moreover, overexpression of *HY5* in soybean has been witnessed to enhance UV-B tolerance and flavonoid content without compromising growth. This trait is considered very important in fields which are exposed to increasing solar radiation [129]. Integrating multi-omics data with machine learning is accelerating gene discovery; a recently reported model has successfully predicted *PIF4* as a key target for improving shade tolerance in densely planted rice paddies. This has been further validated by field phenotyping efficiently [130]. Such innovations may be able to highlight the potential of tailoring light–clock networks to mitigate climate impacts in the near future.

**Table 2 biology-14-01375-t002:** Translational Applications for Light–Clock Engineering. Recent successes in crop engineering highlight the potential of targeting light–clock hubs for climate adaptation.

Crop	Target Gene	Engineering Approach	Outcome	References
Rice	*Ghd7*	CRISPR editing	Photoperiod insensitivity, delayed flowering	[131]
Wheat	*ELF3*	Allele mining	Improved heat tolerance and crop	[132]
Tomato	*HY5*	CRISPR knockout	Enhanced fruit yield under low light	[133]
Barley	*Ppd-H1*	Marker-assisted selection	Early flowering, adaptation to long days	[134]
Maize	*ZmCCT*	CRISPR editing	Delayed flowering, improved biomass	[135]
Soybean	*E1*	RNA interference (RNAi)	Extended flowering period, higher yield	[136]
Potato	*StSP6A*	CRISPR editing	Tuber formation under long-day conditions	[137]
*Arabidopsis*	*TOC1*	CRISPR editing	Altered circadian rhythms for stress adaptation	[42]
Sorghum	*SbPRR37*	CRISPR editing	Delayed flowering, improved drought tolerance	[138]
Sugar beet	*BvBTC1*	CRISPR editing	Bolting resistance, improved root yield	[139]
Grapevine	*VvCO*	Overexpression	Improved berry size and ripening timing	[140]
Strawberry	*FaTFL1*	RNAi	Continuous flowering, higher fruit yield	[141]

### 6.3. Policy and Technological Integration

While translating research achievement into agricultural practices it is required to have interdisciplinary collaborations along with policy support. It has been witnessed that public–private partnerships are significantly advancing CRISPR-edited crops through regulatory frameworks, as we have seen the approval of photoperiod-insensitive *OsPRR37* rice [142]. Moreover, various digital agriculture tools including IoT-enabled smart greenhouses have been using real-time light and temperature data. This will help in the adjustment of circadian-aligned LED spectra, optimizing tomato growth under erratic climates [143]. Policy initiatives such as the EU’s “Farm2Fork” strategy can now prioritize circadian-informed cropping systems for the reduced agrochemical use, leveraging natural pest resistance peaks timed to circadian gating.

## 7. Challenges and Future Directions

Critical knowledge gaps persist in tissue-specific GRNs, non-coding RNA regulation, and evolutionary dynamics of light–clock integration. Translational barriers include genotype-by-environment interactions and regulatory hurdles for gene-edited crops. Interdisciplinary synergies, combining AI-driven prediction models, synthetic biology, and participatory breeding are essential to bridge lab-to-field gaps.

### 7.1. Unresolved Questions in Light–Clock Crosstalk

Despite significant progress, fundamental questions regarding the spatiotemporal dynamics and evolutionary plasticity of light–clock networks remain unresolved. A primary challenge is explaining tissue- and cell-type-specific GRNs. While core mechanisms are well-characterized in model species like *Arabidopsis*, how these networks operate across different tissues such as the distinct circadian rhythms observed in root endodermis versus epidermis is poorly understood [144]. The role of non-coding RNAs (like miRNAs and lncRNAs) in controlling the link between light and the clock is becoming an important topic. For example, recent studies suggest that miR172 and miR156 show daily rhythms and control flowering time by targeting the clock genes *TOC1* and *GI*, but their complete roles are still not known [145].

Another major gap lies in understanding the evolutionary dynamics of these networks. While core components are conserved, their functional interactions have diversified across lineages. For example, the *ELF3* gene has acquired novel roles in thermosensing in eudicots, while in monocots like barley, *ELF3* orthologs primarily regulate photoperiodic flowering [60]. Decoding how these evolutionary innovations arose through gene duplication, neofunctionalization, or changes in protein–protein interactions will provide insights into the adaptive potential of crops [146]. Additionally, the integration of novel photoreceptors from algae or bacteria into plant circadian systems, an area of synthetic biology, could reveal fundamental principles of network evolution and create new opportunities for engineering.

Translating mechanistic information from controlled laboratory environments to unpredictable field conditions presents a formidable challenge. Genotype-by-environment (GxE) interactions often mask the benefits of engineered traits; for example, a *CRY1* overexpression allele that enhances yield in one maize hybrid may reduce it in another due to epistatic interactions. Furthermore, climate change introduces compounding stressors (e.g., drought + heat + high light) that may disrupt synergies within light–clock networks, leading to unexpected phenotypes. Overcoming these barriers requires a more complete understanding of how environmental cues are integrated at the whole-plant level.

### 7.2. Interdisciplinary Synergies for Sustainable Agriculture

It is obvious that future progress mainly hinges through integrating molecular biology, ecology, and data-driven sciences. With the help of artificial intelligence (AI) optimal light–clock gene combinations could be predicted, keeping in view the target environments [147]. Recently, a neural network has been reported to be trained on *Arabidopsis* accessions that has successfully predicted *GI-CO* allelic combinations assumed to maximize yield through varied daylengths and validated in tomato field trials [148]. Similarly, synthetic biology tools like optogenetically controlled *TOC1* circuits have the potential to enable real-time clock adjustments in response to satellite weather forecasts. It is assumed that this system will ultimately be used in optimizing growth in precision agriculture paradigms [149].

Collaborative efforts are essential and the Climate-Resilient Crops Consortium (CRCC), a global initiative, unites plant biologists, climatologists, and farmers to test circadian-informed breeding strategies across 50+ crop species. Such partnerships should be expanded to include policymakers and indigenous communities that could ensure solutions that are equitable and context-specific.

### 7.3. Integrating Multi-Omics with Computational Modeling

The integration of multi-omics data with sophisticated computational modeling represents a critical frontier for deciphering the complex, dynamic interactions between light, circadian rhythms, and stress responses [73]. While transcriptomic studies have successfully mapped rhythmic gene expression, it is increasingly clear that a comprehensive understanding requires moving beyond a single omics layer. Proteomic and metabolomic analyses consistently reveal that the rhythms of proteins and metabolites often deviate from their corresponding mRNA patterns, highlighting the significant roles of post-translational modifications, protein turnover, and metabolic feedback loops in shaping the final physiological output [150]. This multi-layered regulation creates a complex system where transcriptional changes are merely the first step in a cascade of events. To accurately capture this regulatory hierarchy, advanced computational frameworks are essential. These frameworks must integrate high-resolution time-series data from transcriptomics, proteomics, and metabolomics to reconstruct dynamic network models that more accurately represent the flow of information from gene to phenotype, moving from correlation to causation in our understanding of how environmental signals are processed and translated into adaptive growth and resilience strategies [151].

These integrative models are already yielding profound insights. For instance, models that combine protein–protein interaction networks with transcriptomic data have elucidated how light-induced phosphorylation dictates the activity of PIF transcription factors, a key regulatory event that occurs independently of transcriptional changes [152]. Similarly, the integration of metabolomic flux data with transcriptional networks has clarified how the circadian clock gates the expression of flavonoid biosynthesis genes, ensuring UV-B protective compounds are synthesized precisely when needed while avoiding costly metabolic investments during the night [153]. Emerging machine learning approaches, such as weighted gene co-expression network analysis applied to multi-omics datasets, are now identifying previously unknown hub genes that connect circadian timing with drought-responsive pathways [154]. The power of single-cell omics further refines this picture by revealing how these integrated networks operate with distinct dynamics in different cell types, such as the unique circadian rhythms observed in the root endodermis compared to the epidermis, suggesting a level of specialized functional organization that was previously invisible to bulk analysis [155].

Looking forward, the field must prioritize the development of multi-scale models that can span from molecular interactions to whole-plant physiology and performance in the field. The ultimate challenge and goal are to create in silico predictive models that can forecast how genetic perturbations—such as CRISPR edits to *PRR7* or *ELF3*—will influence crop resilience under specific, fluctuating environmental conditions, such as heatwaves combined with altered photoperiods. This ambitious endeavor requires not only the integration of diverse omics data but also the incorporation of environmental sensor data, high-throughput field phenotyping, and detailed information on soil and microclimate. Such a holistic, systems-level approach is indispensable for transforming our mechanistic insights into robust, predictive knowledge.

## 8. Conclusions

The integration of light and circadian signaling within plant gene regulatory networks (GRNs) represents a fundamental adaptive paradigm, enabling sessile organisms to anticipate and respond to dynamic environmental conditions. This review has synthesized the molecular crosstalk between photoreceptor-mediated pathways and the circadian oscillator, highlighting how their convergence fine-tunes essential processes from photomorphogenesis to abiotic stress resilience. The translational potential of this knowledge is immense, offering a roadmap for climate-resilient crop engineering. Emerging technologies are pivotal in this endeavor: CRISPR-based genome editing allows for precise manipulation of key network hubs like *PIFs* and *ELF3*; single-cell omics unveils previously obscured, tissue-specific regulatory dynamics; and artificial intelligence integrates multi-omics data to predict network behavior under complex environmental scenarios. Despite significant progress, critical challenges remain a crucial issue. A deeper understanding of tissue-specific and cell-type-specific regulatory circuits, the influence of natural genetic variation on network plasticity, and the ability to predict phenotype from genotype under fluctuating field conditions are key frontiers. The path forward necessitates the integration of multi-omics data across spatial and temporal scales with advanced computational modeling to construct predictive digital models of plant–environment interactions. Ultimately, by bridging fundamental molecular insights with computational power and field-based validation, and by fostering global, interdisciplinary collaboration, we can fully harness the plasticity of light–clock systems. This integrated approach is not merely an academic pursuit but a critical strategy for ensuring sustainable agriculture and global food security in an era of accelerating climate change.

## Figures and Tables

**Figure 1 biology-14-01375-f001:**
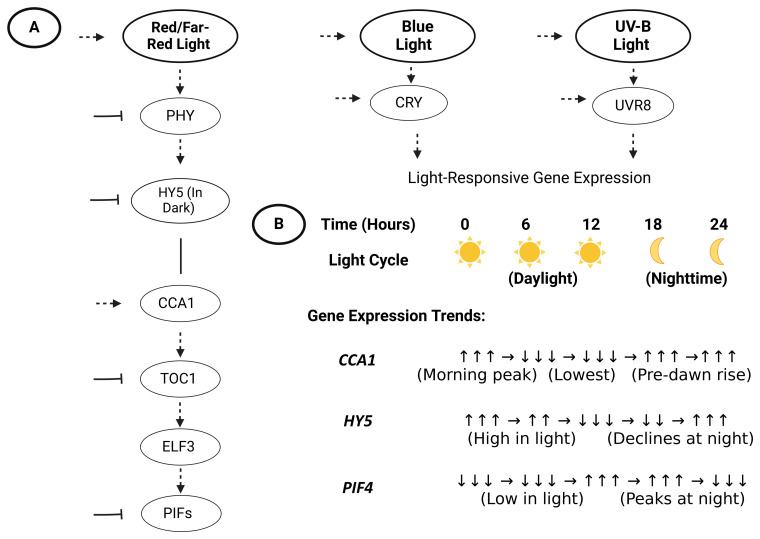
Integration of light and circadian signaling in plant gene regulatory networks. Panel (**A**): Simplified model showing photoreceptors (*PHY, CRY, UVR8*) and their downstream signaling cascades (*COP1/SPA, PIFs, HY5*) interacting with core circadian clock components (*CCA1, TOC1, ELF3*). Arrows indicate activation; blunt lines denote repression. Panel (**B**): Diurnal oscillation of key genes, viz., *CCA1*, *PIF4*, *HY5* under light/dark cycles, showing temporal regulation.

**Figure 2 biology-14-01375-f002:**
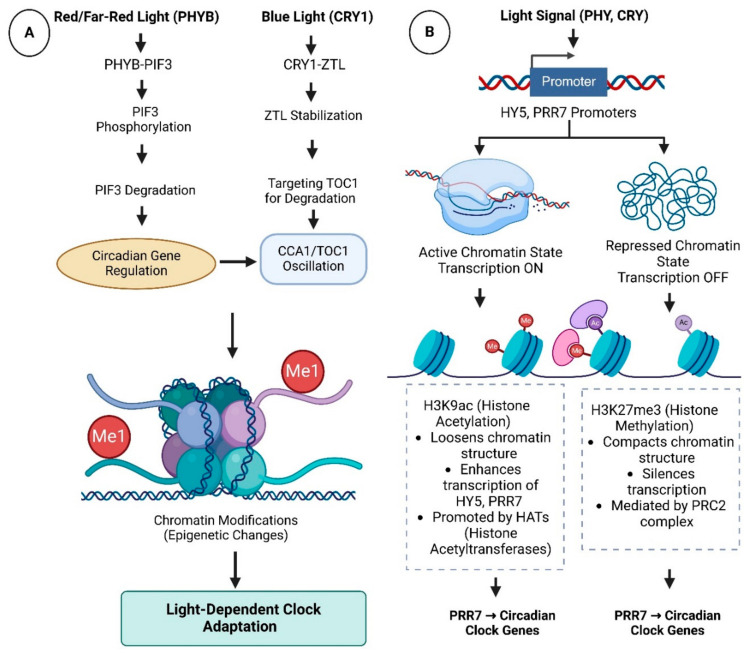
Molecular Mechanisms of Light–Clock Crosstalk; Showing post-Translational and Epigenetic Regulation of Light–Clock Integration; Panel (**A**): Protein–protein interaction networks (e.g., *PHYB-PIF3*, *CRY1-ZTL*) regulating clock resetting and light signaling; Panel (**B**): Elaboration chromatin remodeling at shared promoters (e.g., *HY5*, *PRR7*) via histone acetylation (H3K9ac) and methylation (H3K27me3).

**Figure 3 biology-14-01375-f003:**
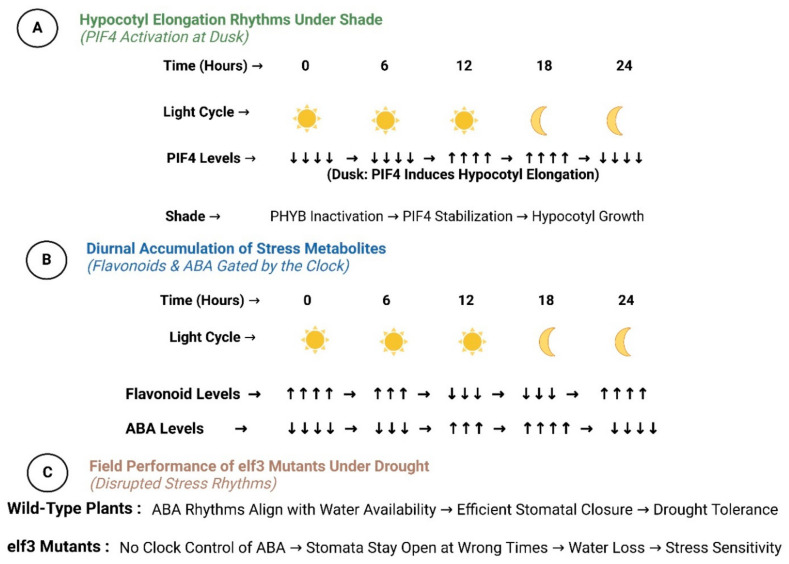
Diurnal Regulation of Photomorphogenesis and Stress Responses; Representing temporal coordination of growth and stress adaptation by Light–Clock networks; Panel (**A**): Hypocotyl elongation rhythms under shade, showing *PIF4* activation at dusk.; Panel (**B**): Diurnal accumulation of stress metabolites (e.g., flavonoids, ABA) gated by the clock; Panel (**C**): Field performance of *elf3* mutants under drought, showing disrupted stress rhythms.

**Figure 4 biology-14-01375-f004:**
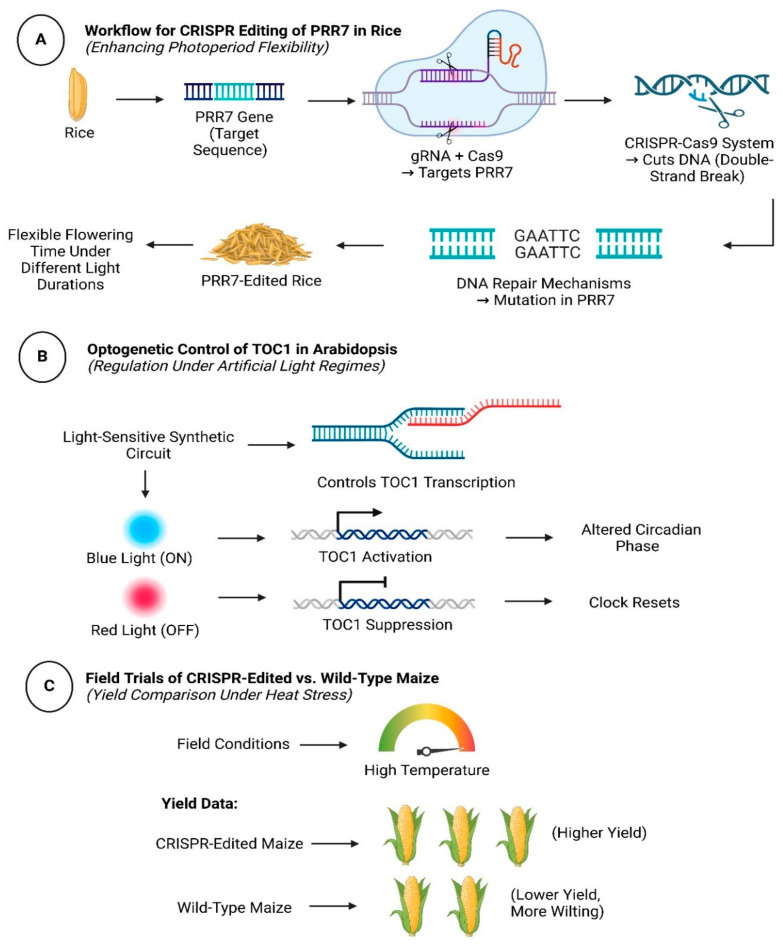
CRISPR-Editing and Synthetic Biology Applications; Showcasing engineering light–clock networks for climate-resilient crops; Panel (**A**): Workflow for CRISPR editing of *PRR7* in rice to enhance photoperiod flexibility; Panel (**B**): Optogenetic control of *TOC1* expression in *Arabidopsis* under artificial light regimes; Panel (**C**): Field trial results comparing yields of CRISPR-edited vs. wild-type maize under heat stress.

**Table 1 biology-14-01375-t001:** Core components of light–clock networks exhibit conserved roles across plant lineages, with mutations often leading to developmental or circadian defects.

Component	Function	Mutant Phenotype	Evolutionary Conservation	Ref.
*PHYB*	Red light sensing	Elongated hypocotyl	*PHY* homologs in moss	[41]
*CCA1*	Morning oscillator	Arrhythmic growth	*CCA1* homologs in rice	[42]
*HY5*	Light-responsive transcription factor	Reduced photomorphogenesis	*HY5* homologs in algae and higher plants	[25]
*LHY*	Morning oscillator	Arrhythmic growth	*LHY* homologs in angiosperms	[42]
*TOC1*	Evening oscillator	Altered circadian rhythms	*TOC1* homologs in Arabidopsis and rice	[2]
*CRY1*	Blue light sensing	Hypersensitive to blue light	*CRY* homologs in algae and higher plants	[43]
*CRY2*	Blue light sensing	Delayed flowering	*CRY* homologs in algae and higher plants	[43]
*ZTL*	Circadian clock regulator	Altered circadian period	*ZTL* homologs in angiosperms	[44]
*PRR7*	Circadian clock regulator	Altered circadian rhythms	*PRR7* homologs in *Arabidopsis* and rice	[45]
*PRR9*	Circadian clock regulator	Altered circadian rhythms	*PRR9* homologs in *Arabidopsis* and rice	[45]
*ELF3*	Light input to clock	Early flowering, arrhythmic growth	*ELF3* homologs in monocots and dicots	[46]
*ELF4*	Light input to clock	Altered circadian rhythms	*ELF4* homologs in monocots and dicots	[31]
*PIF4*	Light signaling and growth	Reduced hypocotyl elongation	*PIF4* homologs in angiosperms	[47]
*PIF5*	Light signaling and growth	Reduced hypocotyl elongation	*PIF5* homologs in angiosperms	[48]

## Data Availability

The data supporting this review are from previously reported studies and datasets, which have been cited in the manuscript.

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
