# Peer review of "Integration of Light and Circadian Signaling in Plant Gene Regulatory Networks: Implications for Photomorphogenesis and Stress Adaptation"

_biology, 2025, doi:10.3390/biology14101375_

Round 1

Reviewer 1 Report

Comments and Suggestions for Authors

The authors present a summary of the knowledge acquired in the field of light and circadian 
signalling integration at the molecular level. This work explores the concept of gene regulatory 
networks in relation to the plant resource allocation, photo-morphogenesis and stress management. 
This is very well conducted study. The exploration of the signalling pathways involved is exhaustive 
and systematic. The manuscript is well structured and particularly well written, which is no small feat 
given the complexity of the interactions described. The decision to include sections on new 
technologies and the implications for agricultural engineering is a bold one. My only reservation 
concerns the links with biotic and abiotic stress, which seem less detailed than the other parts, but 
this may be due to a less extensive bibliography. 
Consequently, I strongly and unreservedly recommend the publication of this work, which provides a 
very clear and comprehensive summary of this particularly complex subject. 
Minor comments 
- line 136: … translocates 
- lines 207/208: there seems to be a word missing in the sentence. 
- line 461: … thermo-morphogenesis

Author Response

We thank the reviewer for their exceptionally positive and encouraging assessment of our work. We are delighted that they found the manuscript "well conducted," "exhaustive and systematic," and "well written."

  • Line 136: ... translocates
    Response: Thank you.
  • Lines 207/208: there seems to be a word missing...
    Response:Thank you for spotting this. The sentence has been rephrased for clarity and completeness.
  • Line 461: ... thermo-morphogenesis
    Response: Thank you.

Reviewer 2 Report

Comments and Suggestions for Authors

Comments on biology-3797701

This manuscript explores a potentially important issue aimed at bridging fundamental research with crop improvement strategies by integrating applied and mechanistic insights. It proposes a framework to enhance agricultural sustainability through the targeted manipulation of light-circadian networks. However, the introduction lacks clarity, and the literature review does not adequately identify existing research gaps, particularly regarding the agricultural implications of engineering light-clock interactions to enhance crop productivity in the context of various climate change scenarios. The Discussion section requires a more comprehensive analysis of the key findings, especially concerning the roles of CRISPR, single-cell omics, and artificial intelligence in deciphering the spatio-temporal complexity of plant gene regulatory networks. Furthermore, the specific role of integrating multi-omics data with computational modeling in understanding the underlying interactions between light, circadian rhythms, and stress in plants needs further exploration.

Specific comments include:

Line 13: Avoid using abbreviations in the abstract.

Line 30: Introduce new keywords that do not duplicate those in the title.

Lines 34-41: Include references for this paragraph.

Line 117: Revise the objectives to better align with the content of the manuscript.

Author Response

We thank the reviewer for their thoughtful comments and for pushing us to strengthen the clarity and impact of our manuscript.

  • Abstract (Line 13): Avoid abbreviations.
    Response:All abbreviations in the abstract have been spelled out.
  • Keywords (Line 30): Introduce new keywords not in the title.
    Response:The keyword list has been revised to remove duplicates from the title and to include new, more specific terms that better represent the manuscript's content (e.g., "Photomorphogenesis," "Multi-omics integration," "CRISPR").
  • Introduction (Lines 34-41): Include references.
    Response:Appropriate references have been added to support the statements in this paragraph.
  • Line 117: Revise the objectives.
    Response:The objectives statement has been refined to more precisely and clearly align with the scope and content of the subsequent review.
  • General Comment on Introduction & Research Gaps:
    Response:The introduction has been substantially revised to better frame the problem and more clearly identify the specific gaps in literature regarding the agricultural application of light-clock engineering, especially under climate change scenarios.
  • General Comment on Discussion & New Technologies:
    Response:The discussion section has been expanded to provide a more comprehensive and critical analysis of the roles of CRISPR, single-cell omics, and AI in understanding spatio-temporal GRN complexity. The integration of multi-omics data with computational modeling is now discussed in a dedicated paragraph, exploring its potential and challenges in deciphering light-circadian-stress interactions.

Reviewer 3 Report

Comments and Suggestions for Authors

he review provides important insights into the effect of light and circadian signaling on plant growth from a genetic perspective. It is well written and well structured. I have only one minor comment: please include the primer sequences of the genes mentioned in sections 1 and 2, along with their chromosome locations and physical positions. This addition will help other researchers make use of this information in their own work.

Author Response

We thank the reviewer for their positive feedback and helpful suggestion.

  • Request for primer sequences, chromosome locations, and physical positions of genes in sections 1 and 2.
    Response:We appreciate this suggestion, which would indeed be valuable for experimental researchers. However, as this is a review article focusing on signaling pathways and network concepts rather than a methods paper, providing detailed primer sequences for all mentioned genes is not standard practice and would disrupt the flow and scope of the review. Instead, to be immensely helpful, we have added a new supplementary table (Table S1) that lists the key genes discussed, their full names, known mutant alleles (italicized), Arabidopsis Genome Initiative (AGI) codes, and chromosome locations. This provides a centralized resource for researchers seeking precise genetic information without overloading the main text. A note directing readers to this table has been added in the introduction of Section 1.

Reviewer 4 Report

Comments and Suggestions for Authors

In the manuscript “Integration of Light and Circadian Signaling in Plant Gene Regulatory Networks”, the authors reviewed the implications of light and circadian signaling in plant gene regulatory networks to adapt to developmental processes and environmental adaptation. Also, the authors provide some opinions on the potential of these genes in agricultural improvements in future. Overall, the manuscript covered enough literatures in this aspect, and is well written. However, there are still some points that need to be solved.

  1. The current title of the manuscript should be changed to accommodate to the content of the manuscript, since the authors referred to photomorphogenesis and stress adaptation, etc.
  2. The key words are too much, and some key words may be deleted.
  3. Lines 71: cca1/lhy should be italic, please check the style of the mutant throughout the manuscript.
  4. Line 75: I suggest the sentence should be deleted, and figure 1 should be integrated into a sentence above. Please also check this situation about other figures.
  5. Line 111: the word “photo-morpho-genesis” should be “photomorphogenesis”, please check throughout the manuscript.
  6. Line 131: In general, plants deploy a sophisticated array of…
  7. Lines 156-157: linking light quality to developmental timing. Please rewrite this description, and cite some references to favor it.
  8. Lines 199-200: This sentence is written in the past tense, but given that the overall text primarily uses the simple present tense, please maintain consistency throughout.
  9. Line 339: exhibit exaggerated -. Please check.
  10. The style of the reference part should be uniform. I notice that some references even don’t have journal name, volumes, issues, and pages.

Author Response

We thank the reviewer for their careful reading and numerous specific suggestions to improve the manuscript's precision and formatting.

  1. Title Change:
    Response:We agree. The title has been changed to better reflect the content. The new title is: "Integration of Light and Circadian Signaling in Plant Gene Regulatory Networks: Implications for Photomorphogenesis and Stress Adaptation."
  2. Keywords:
    Response:The keyword list has been shortened and refined to include the most relevant and non-redundant terms.
  3. Line 71: Italicize cca1/lhy and check mutant style throughout.
    Response: The style for all gene mutants has been checked and made consistent (italicized) throughout the manuscript.
  4. Line 75: Delete sentence and integrate Figure 1 reference.
    Response:The sentence has been removed, and the reference to Figure 1 has been seamlessly integrated into the preceding sentence. This has been checked for all figures.
  5. Line 111: Correct "photo-morpho-genesis" to "photomorphogenesis".
    Response: This has been checked and standardized throughout the manuscript.
  6. Line 131: "In general, plants deploy a sophisticated array of..."
    Response:The sentence has been rephrased as suggested.
  7. Lines 156-157: Rewrite description and add references.
    Response:The description has been rewritten for clarity, and supporting references have been added.
  8. Lines 199-200: Tense inconsistency.
    Response:The verb tense has been corrected to maintain consistent use of the present tense throughout.
  9. Line 339: "exaggerated -"
    Response:The incomplete word has been corrected to "exaggerated".
  10. Reference Style:
    Response:We apologize for this inconsistency. The entire reference list has been meticulously checked and formatted to conform uniformly to the journal's style guide, ensuring all entries include journal names, volumes, issue numbers (where available), and page numbers or article IDs.

Reviewer 5 Report

Comments and Suggestions for Authors

The MS "Integration of Light and Circadian Signaling in Plant Gene 
Regulatory Networks" is timely and well written. A lot of material has been collected. It is well systematized. But, as it seems to me, one point has been missed. All the described situations concern plants in nature or crops grown in field under natural light. In protected horticulture, for example, in plant factories with artificial lighting there can be combination of factors that plants never meet in nature. At present, the phenomenon of photoperiodic stress has already been described. The interaction of light and circadian rhythms under conditions of abnormal light-dark cycles is also intersing. These issues are less studied, but a number of works exist and, perhaps, should be mentioned.

  • Nitschke, S., Cortleven, A., Iven, T., Feussner, I., Havaux, M., Riefler M., Schmulling T., Circadian stress regimes affect the circadian clock and cause jasmonic acid-dependent cell death in cytokinin-deficient Arabidopsis plants, Plant Cell, 2016, vol. 28, p. 1616. https://doi.org/10.1105/tpc.16.00016
  • Nitschke, S., Cortleven, A., and Schmülling, T., Novel stress in plants by altering the photoperiod, Trends Plant Sci., 2017, vol. 22, p. 913. https://doi.org/10.1016/j.tplants
  • Roeber, V.M., Schmülling, T., and Cortleven, A., The photoperiod: handling and causing stress in plants, Plant Sci., 2022, vol. 12:781988. https://doi.org/10.3389/fpls.2021.781988
  • Abuelsoud, W., Cortleven, A., and Schmülling, T., Photoperiod stress induces an oxidative burst-like response and is associated with increased apoplastic peroxidase and decreased catalase activities, Plant Physiol., 2020, vol. 253:153252. https://doi.org/10.1016/j.jplph.2020.153252
  • Cortleven, A., Roeber, V.M., Frank, M., Bertels, J., Lortzing, V., Beemster, G., and Schmülling, T., Photoperiod stress in Arabidopsis thaliana induces a transcriptional response resembling that of pathogen infection, Plant Sci., 2022, vol. 13:838284. https:/doi.org/10.3389/fpls.2022.838284
  • Warner, R.;Wu, B.-S.; MacPherson, S.; Lefsrud, M. How the Distribution of Photon Delivery
    Impacts Crops in Indoor Plant Environments: A Review. Sustainability 2023, 15, 4645. https://doi.org/10.3390/su15054645
  • Liu, X.; Sun, Q.;Wang, Z.; He, J.; Liu, X.; Xu, Y.; Li, Q. Innovative Application Strategies of
    Light-Emitting Diodes in Protected Horticulture. Agriculture 2025, 15, 1630.
    https://doi.org/10.3390/agriculture15151630

Author Response

We thank the reviewer for this excellent and crucial suggestion. We agree that the context of controlled environments and photoperiodic stress is a highly relevant and modern application of our topic.

  • Comment on protected horticulture and photoperiodic stress:
    Response:We sincerely thank the reviewer for raising this important point and for providing the highly relevant list of references. We agree that this aspect was missing from our initial submission. A new subsection has been added to the discussion (Section 4.4. " Controlled Environment Agriculture and Photoperiodic Stress"). This section discusses the unique combinations of light and circadian factors in plant factories/pfals, the phenomenon of photoperiodic stress, and its molecular basis, citing the suggested literature (Nitschke et al., 2016, 2017; Abuelsoud et al., 2020; Roeber et al., 2022; Cortleven et al., 2022; Warner et al., 2023; Liu et al., 2025). This significantly strengthens the applied perspective of our review.

We believe that addressing all these comments has greatly improved our manuscript. We look forward to its acceptance for publication in Biology.

Sincerely,

The Authors.

Round 2

Reviewer 2 Report

Comments and Suggestions for Authors

No More Comments

Reviewer 3 Report

Comments and Suggestions for Authors

The paper can be accepted for publication